# Gastrointestinal Microbiome and Neurologic Injury

**DOI:** 10.3390/biomedicines10020500

**Published:** 2022-02-21

**Authors:** Eric J. Panther, William Dodd, Alec Clark, Brandon Lucke-Wold

**Affiliations:** 1Department of Neurosurgery, University of Florida, Gainesville, FL 32601, USA; brandon.lucke-wold@neurosurgery.ufl.edu; 2College of Medicine, University of Central Florida, Orlando, FL 32816, USA; wsdodd@ufl.edu (W.D.); acclark@knights.ucf.edu (A.C.)

**Keywords:** gut microbiome, neurologic injury, enteric nervous system, emerging approaches

## Abstract

Communication between the enteric nervous system (ENS) of the gastrointestinal (GI) tract and the central nervous system (CNS) is vital for maintaining systemic homeostasis. Intrinsic and extrinsic neurological inputs of the gut regulate blood flow, peristalsis, hormone release, and immunological function. The health of the gut microbiome plays a vital role in regulating the overall function and well-being of the individual. Microbes release short-chain fatty acids (SCFAs) that regulate G-protein-coupled receptors to mediate hormone release, neurotransmitter release (i.e., serotonin, dopamine, noradrenaline, γ-aminobutyric acid (GABA), acetylcholine, and histamine), and regulate inflammation and mood. Further gaseous factors (i.e., nitric oxide) are important in regulating inflammation and have a response in injury. Neurologic injuries such as ischemic stroke, spinal cord injury, traumatic brain injury, and hemorrhagic cerebrovascular lesions can all lead to gut dysbiosis. Additionally, unfavorable alterations in the composition of the microbiota may be associated with increased risk for these neurologic injuries due to increased proinflammatory molecules and clotting factors. Interventions such as probiotics, fecal microbiota transplantation, and oral SCFAs have been shown to stabilize and improve the composition of the microbiome. However, the effect this has on neurologic injury prevention and recovery has not been studied extensively. The purpose of this review is to elaborate on the complex relationship between the nervous system and the microbiome and to report how neurologic injury modulates the status of the microbiome. Finally, we will propose various interventions that may be beneficial in the recovery from neurologic injury.

## 1. Introduction

An intricate communication between the gastrointestinal (GI) tract’s enteric nervous system (ENS) and the central nervous system (CNS) creates a unique dynamic unlike any other peripheral organ system. Together, the GI tract’s intrinsic and extrinsic neurologic inputs influence its movement patterns, blood flow, reflexes, and interactions with the gut immune and endocrine systems [1]. While the GI tract’s intrinsic neural plexuses allow the system a degree of autonomy in executing many of these functions, the CNS plays an integral role in regulating and modulating these in response to external stimuli [1,2].

Extrinsic neuronal communication with the GI tract occurs via vagal, spinal thoracolumbar, and spinal lumbosacral innervation [1,2]. Vagal efferents arising from the dorsal motor nucleus (DMN) consist of both excitatory and inhibitory lower motor neurons (LMNs) as well as preganglionic parasympathetic fibers [3]. Vagal influence on the GI tract is most prominent in the esophagus and stomach, where responsibilities include upper esophageal sphincter (UES) contraction, striated and smooth muscle peristalsis, and regulation of hormonal release [1,3]. Without vagal efferents, the upper and lower esophagus can no longer propel its contents forward [4,5]. Vagal sensory neurons, or afferents, at the level of the esophagus, stomach, and proximal small intestine communicate with the CNS to mediate numerous vasovagal reflexes in addition to sensing satiety [6,7]. Similar to vagal innervation in the upper GI tract, thoracolumbar innervation on the middle GI tract consists of both efferents and afferents. Thoracolumbar preganglionic sympathetic efferents innervate their postganglionic counterparts in the celiac, superior mesenteric, and inferior mesenteric ganglia. In addition to their significant influence on GI tract vasculature, these fibers function in slowing transit times of tract contents directly via sphincter contraction and indirectly via inhibition of myenteric and submucosal ganglia [1,2,8]. Thoracolumbar afferents compromise the majority of thoracolumbar innervation in the GI tract. Although primarily inactive in nonpathological states, sensitization of thoracolumbar afferents via gut inflammation plays a role in pain sensation [1,6]. Lumbosacral input on the GI tract is primarily in the form of parasympathetics which innervate their respective postganglionic cell bodies in the pelvic plexus or act indirectly via the ENS myenteric plexus [2,9]. Similar to vagal parasympathetics in the upper GI tract, lumbosacral parasympathetic efferents provide excitatory and inhibitory innervation to the distal colon to increase or decrease motility, respectively [2,10]. Lumbosacral afferents communicate stretch and pain to the CNS, namely Barrington’s nucleus [2,11,12,13]. Lumbosacral sensory and motor neurons also function in important lower GI reflexes, such as defecation [1,14,15].

Despite the importance of CNS innervation in proper digestive system functioning described above, the ENS gives the GI tract the ability to maintain many of its functions independent of extrinsic support [16,17,18,19]. The ENS consists of approximately 20 neuronal subtypes dispersed in its two major ganglia, the myenteric extending from esophagus to anus and submucosal in the small and large intestines [1]. In the esophagus, nitric oxide producing enteric neurons allow for sphincter relaxation independent of vagal inhibition [1,20]. ENS innervation in the stomach is responsible for gastric acid secretion through its direct innervation of gastrin-releasing G cells [5,20,21]. In the small and large intestines, enteric neurons function in fluid movement and balance; blood flow; nutrient handling; gut-wall integrity; and communicating with local and peripheral neural, endocrine, and immune cells [1,5,22]. Through intrinsic sensory neurons, interneurons, and motor neurons, the ENS is responsible for controlling small intestine motility and propulsion [5,23,24]. Likewise, the migrating motor complex (MMC), a small intestinal phenomenon important for preventing bacterial overgrowth, is dependent entirely on ENS neurons [25,26]. Enteric nociceptive neurons are important for retropulsive reflexes such as vomiting in the small intestine and for propulsive contractions and copious fluid secretion in the colon [5,27,28]. With the support of sympathetic pathways, small intestinal secretomotor neurons regulate fluid movement and electrolyte secretion between the intestinal lumen and body fluid compartments [1,29,30,31,32,33]. In the colon, the ENS is capable of reproducing the defecation reflex with lumbosacral stimulation independent of central command [34].

## 2. Neurologic Control of the Gut Microbiome

The gut microbiome contains trillions of bacteria, viruses, and fungi that are critical for the health of the organism. The majority of these microbes are symbiotic; however, pathogenic bacteria can invade the gut and lead to diseases such as cancer, autoimmunity, and multiple sclerosis [35]. Thus, tight neuronal control of this system is critical in order to maintain homeostasis and prevent disease. This control is achieved through intrinsic (enteric) and extrinsic innervation of the gut.

### 2.1. Intrinsic (Enteric) Nervous System

Intrinsic, or enteric, neurons function to regulate the motility, secretion, and immunologic defense of the gut largely independent of CNS control [36]. There are nearly 600 million enteric neurons within the gastrointestinal (GI) smooth muscle stemming from the myenteric and submucosal plexuses [37,38]. These neurons communicate with enteric glial cells to control the enteroendocrine cells on the epithelial lining which are responsible for secreting peptide hormones that regulate GI inflammation, secretion, and motility [38,39]. One such hormone is glucagon-like peptide 2 (GLP-2) which acts to decrease intestinal inflammation [40]. Within the ENS there are afferent neurons, interneurons, and motoneurons [41]. Afferent neurons, or intrinsic primary afferent neurons (IPANs), are responsible for relaying stimuli from the gut to the ENS [42]. IPANs relay information to the interneurons. Interneurons of the ENS are subdivided into ascending and descending interneurons [43]. Ascending interneurons are those that are projected orally and release acetylcholine (Ach), and descending interneurons are those that are projected anally and are grouped into three classes based on the signaling molecule they produce/release. Descending interneurons can release (1) acetylcholine, nitric oxide (NO), and vasoactive intestinal peptide (VIP); (2) acetylcholine and somatostatin; or (3) acetylcholine and serotonin [44,45]. Interneurons then pass the signal to motoneurons which function to innervate the musculature of the GI tract. There are excitatory motoneurons which secrete Ach and substance P (SP) and inhibitory motoneurons which secrete NO and VIP. Through muscular innervation, these neurons direct GI motility from mouth to anus (Figure 1).

### 2.2. Microbiome Effect on Enteric Nervous System

The microbes that make up the gut microbiome are capable of releasing short-chain fatty acids (SCFAs), neurotransmitters, gaseous factors, and lipopolysaccharides that have an effect on the functions of the ENS [41]. Lipopolysaccharides have been shown to act on toll-like receptors (TLRs) 2 and 9 in the ENS, leading to anti-inflammatory effects [46]. Similarly, SCFAs are natural byproducts of microbial metabolism which have been shown to bind G-protein-coupled receptors (GPCRs) located on the enteroendocrine cells, leading to hormone modulation and motility effects [47]. Additionally, there are many strains of bacteria that are known to release neurotransmitters serotonin, dopamine, noradrenaline, γ-aminobutyric acid (GABA), acetylcholine, and histamine [48]. These neurotransmitters have a wide effect including anti-inflammation through histamine, impacts on mood and behavior through serotonin and tryptamine, and increases in motility and gastric emptying through GABA.

### 2.3. Extrinsic Innervation of Gut

Extrinsic innervation of the gut describes communication from the brain to the gut (brain–gut axis) through autonomic neurons and from the gut to the brain (gut–brain axis) through somatosensory neurons. The extrinsic somatosensory neurons contain nerve endings in the gut that project into the central nervous system [41]. This allows a connection between the gut and CNS that gives information about the condition of the gut. The connection is achieved through vagal and spinal pathways [49]. Cholecystokinin (CCK) is another major mediator of gastrointestinal feedback to the central nervous system through the afferent component of the vagus nerve [37].

As previously mentioned, the gut microbiota is capable of consuming and releasing neurotransmitters such as γ-aminobutyric acid (GABA), serotonin, glutamate, dopamine, and norepinephrine [50]. It has been shown that the presence of GABA-producing bacteria can lead to depression, showing a relationship between the gut microbiome and the CNS [51].

## 3. Mechanisms of Microbiome Disruption in Neurologic Disease

The brain–gut axis [52] is a well-characterized, multidirectional interaction between the gastrointestinal, immune, and nervous systems. Injury, disease, or other perturbation of these systems affects the function of the others [53,54]. For example, activation of certain neuronal circuits can increase immune response to bacterial infection, and microbiota-depleted mice display altered behavioral patterns and CNS structure [55,56,57]. The precise mechanisms by which these systems interact in health and disease are still under investigation. The leading hypotheses suggest immune cell education and development in the gut and CNS alter trafficking patterns and proinflammatory pathway activation after injury. Immune cells, which are constantly surveilling both the gut and CNS, respond to neurotransmitters, providing a straightforward mechanism for neuronal activity to alter immune cell function [58,59,60].

Another possible mechanism is the function of gut microbiota-derived metabolites. Microbiota depletion disrupts microglia development and function; however, treatment with microbiota-derived short-chain fatty acids (SCFAs) can restore these phenotypes. Further, knockout of an SCFA receptor causes microglia phenotypes similar to those seen in microbiota-depleted mice [61]. Other gut flora-associated metabolites, particularly those of tryptophan, are associated with CNS regulation, possibly acting through aryl hydrocarbon receptors [62,63,64]. In the next sections, we will review how these mechanisms alter the brain–gut axis during specific neurological disease processes.

A comprehensive search of the literature was conducted using PubMed.gov (accessed on 23 January 2022) with the most common search terms of “microbiome or microbiota”, “neurologic injury”, “ischemic stroke”, “spinal cord injury”, “traumatic brain injury”, and “hemorrhagic cerebrovascular lesions”.

### 3.1. Ischemic Stroke

The contribution of the gut microbiota to ischemic stroke is unique in that it affects both risk and outcome. Cross-sectional clinical studies indicate that patients with the most known risk factors for ischemic stroke have significantly altered microbiota composition [65]. Further, high-risk patients have decreased butyrate-producing bacteria and lower fecal butyrate concentrations. Other human studies evaluating outcome after ischemic stroke also find decreased SCFA concentrations in stroke patients compared to healthy controls [66]. Fecal SCFA concentration is also inversely associated with functional outcome at 90 days poststroke. However, it is important to note that these changes could be an epiphenomenon of the disease rather than causative. Poor diet and mobilizability after ischemic stroke could lead to poor microbiome health. Nonetheless, these data suggest that not only is the brain–gut axis a relevant factor in stroke risk and outcome, but metabolites such as SCFAs could be the functional mechanism in this relationship.

One of the primary differences between the microbiota of young and aged mice is the decrease in SCFA-producing bacteria over time [67]. When aged mice receive fecal transplant from young mice after being subjected to middle cerebral artery occlusion, they exhibit better functional recovery than those that received fecal transplant from older aged mice [68]. Changes to specific bacteria genera are also consistent with the observation that different experimental antibiotic regiments have different effects on stroke outcome. Treating mice with amoxicillin and clavulanic acid (*Augmentin*) reduces infarct volume after stroke, while the broad-spectrum cocktail of ampicillin, ciprofloxacin, metronidazole, vancomycin, and imipenem reduces survival [69,70]. These studies indicate that it is not simply the presence, absence, or total burden of gut microorganisms that regulate the gut–brain axis, but rather the relative abundances of and interactions between clusters of bacterial groups.

### 3.2. Spinal Cord Injury

The gut microbiota composition is significantly altered after spinal cord injury in humans [71]. The severity of the spinal cord lesion also predicts the severity of the subsequent gut dysbiosis observed [72]. As with stroke, the specific reduction in SCFA-producing bacteria could be particularly harmful [71,72]. These findings suggest the “gut–brain axis” may actually reach beyond the cranium and affect the entire CNS.

The precise mechanisms of how gut microbiota can affect spinal cord injury (SCI) pathology remain unclear, but there is clearly a functional role for the microbiota in disease progression. Mice, like humans, develop gut dysbiosis after SCI, which can be reduced with fecal transplant [73]. Fecal transplant and probiotic treatment also improve some parameters of behavioral and functional outcome after SCI in mice, suggesting that gut dysbiosis does exacerbate the pathophysiological process [73,74]. On the other hand, a broad-spectrum depletion of microorganisms worsens recovery [74]. A more targeted approach to microbiota manipulation could further elucidate the mechanisms by which the microbiota regulates SCI pathology.

### 3.3. Traumatic Brain Injury

As in ischemic stroke, the brain–gut axis is functionally bidirectional after traumatic brain injury (TBI); in recent mouse studies, neurological injury appears to induce gut dysbiosis which in turn aggravates neuroinflammation and worsens outcome [54,75]. Gut-dysbiosis-induced neuroinflammation seems to be at least partially mediated by microglia and astrocytes, which have also been shown to be regulated by enteric metabolites [54,61,64]. Restoration of healthy microbiota can interrupt this process and improve neurological deficits after TBI [76]. Consistent with the general model of the gut–brain axis, SCFA metabolism seems to be a critical element of TBI-induced gut dysbiosis. The abundance of SCFA-producing bacteria is diminished after TBI, and SCFA supplementation is sufficient to improve neurological function [77].

The role of the gut microbiota after TBI is distinguished from that of the other previously discussed neurological injuries in several ways. First, broad-spectrum antibiotic treatment seems to be neuroprotective after TBI [78]. Similar antibiotic regimens used in models of ischemic stroke and SCI worsen histologic and/or behavioral pathology [70,74]. Another unique element of TBI pathology is that its relationship with the gut microbiota is dependent on the mechanism of injury. While most studies use a model of single, severe cortical impact, others use mild, repetitive TBI as a model of sports- or military-related injury. When mice are subjected to this model of TBI, their microbiota is minimally altered [79]. Further, the neurological deficits seem to be independent of changes to gut microbiota. These studies indicate that while ischemic stroke, SCI, and TBI are pathologically similar in some respects, there are important differences that affect their influence on the gut–brain axis.

### 3.4. Hemorrhagic Cerebrovascular Lesions

Comparatively little is understood about the relationship between hemorrhagic stroke-causing vascular lesions and the gut microbiota. One of the early studies in this field demonstrated certain species of bacteria increase cerebral cavernous malformation (CCM) growth in mice through activating TLR4 [80]. Further, the same report found that germ-free mice raised in sterile conditions have lower incidence of CCM. These findings were correlated to human CCM patients who were found to have polymorphisms promoting TLR4 expression [80]. Intracerebral hemorrhage (ICH), which can be caused by many lesions, including CCMs [81], is also associated with gut dysbiosis [82]. Fecal transplant is capable of improving neurological outcome after ICH [82], though the exact bacterial populations responsible for this effect remain uncertain. To our knowledge, SCFA metabolism has not been evaluated after ICH in humans, but a high plasma concentration of another microbial metabolite, trimethylamine-n-oxide, has been shown to correlate with poor outcome [83]. There has also been at least one case reported of hemorrhage-associated multiorgan system failure being successfully treated with fecal transplant from healthy donors [84]. Collectively, these studies are a strong foundation for further investigation into gut dysbiosis before and after hemorrhagic stroke.

Another type of hemorrhagic stroke, spontaneous subarachnoid hemorrhage (SAH), is primarily caused by intracranial aneurysms [85,86]. There is some evidence that the gut microbiota influences aneurysm formation; depletion of gut bacteria through orally administered broad-spectrum antibiotics reduces aneurysm formation in mice [87]. The mechanism could be similar to other diseases, wherein immune cells are differentially regulated in the gut based on the local microbial status and then circulate to the CNS to mediate disease progression. These studies are intriguing investigations into the formation of SAH-producing lesions, but the function of the gut–brain axis after SAH remains elusive.

## 4. Interventions

Few human clinical studies evaluating the efficacy of interventions targeting the gut–brain axis have been published [88,89]. Thus, the feasibility of such interventions in the clinical setting must be mainly derived from preclinical studies utilizing animal models. The interventions gaining the most traction in this context are the administration of probiotics/prebiotics and fecal microbiota transplantation (FMT). The administration of oral short-chain fatty acids (SCFAs) has shown some promise as well. Other novel techniques in the early stages of investigation will be also discussed.

A comprehensive search of the literature was conducted using PubMed.gov (accessed on 23 January 2022) with the most common search terms of “microbiome OR microbiota”, “neurologic OR brain”, “probiotics OR prebiotics”, “fecal microbiota transplantation”, and “short-chain fatty acids”.

### 4.1. Probiotics/Prebiotics

As it relates to therapeutic modalities targeted at manipulating the gastrointestinal microbiome, the administration of probiotics has arguably shown the most promise to date as an adjunct in symptomatic alleviation of several neurologic disease processes [90,91,92]. It is also the only relevant emerging intervention whose efficacy in neurologic injury patients has been evaluated in the clinical setting with human participants [93,94]. Research involving the use of probiotics predominates over the use of prebiotics in this sphere; nonetheless, some research has been done on this subject using prebiotics and is included in this section for organizational purposes.

Several preclinical studies have now demonstrated the efficacy of the administration of several different species of probiotic bacteria, including lactobacilli and butyrate-producing gut bacteria, on animal neurologic injury models [95,96]. Sun et al. [97] treated mice with intragastric *Clostridium butyricum* (*C. butyricum*) for 2 weeks before subjecting the mice to cerebral ischemia-reperfusion injury. Results showed that the pretreated mice displayed decreased expression of caspase-3 and Bax, suggesting antiapoptotic mechanisms of *C. butyricum*, along with improved neurologic deficits. In another study [98], where mice subjected to TBI were treated with *C. butyricum* for 2 weeks pre-TBI and 2 weeks post-TBI, it was shown that *C. butyricum* increased Bcl-2 and decreased Bax levels, demonstrating similar antiapoptotic effects. Results also showed improved neurologic function and reduced cerebral edema in mice treated with *C. butyricum* compared to TBI controls. In a 2016 study [97], diabetic mice treated with *C. butyricum* also demonstrated decreased caspase-3 levels, as well as increased p-Akt levels, suggesting antiapoptotic effects on neurons. In another study [99], mice subjected to TBI and given *Lactobacillus acidophilus* (*L. acidophilus*) were shown to have a reduction in inflammatory markers, including TNF-α and IL1-β, when compared to TBI mice that did not receive a probiotic. Results also showed that *L. acidophilus* administration was able to restore microbiota composition post-TBI and normalize the numbers of activated and total microglia and astrocytes. Similarly, Akhoundzadeh et al. [100] discovered significantly decreased TNF-α levels (*p* = 0.004) in TBI mice pretreated with probiotics for 2 weeks, along with significantly reduced infarct size (*p* = 0.001), compared to controls.

Furthermore, several human clinical studies have now been published that evaluate the efficacy of probiotics/prebiotics on neurologic injury (primarily TBI) patient outcomes [101]. A 2004 RCT [102] examined 20 TBI patients in the ICU randomized into either a control group, receiving only early enteral feeding, or an intervention group, receiving early enteral feeding plus probiotics. Results indicated that the probiotics group had a significantly lower incidence of infection (*p* = 0.03), shorter critical care unit stay (*p* < 0.01), and fewer days of mechanical ventilation (*p* = 0.04) than the control group. A 2011 single-blind RCT [103] evaluated the use of probiotics on outcomes of 52 severe TBI patients, equally randomized into control and probiotic groups. Researchers noted a decreased incidence of nosocomial infections in the probiotic group, as well as shorter ICU stays and reduction in interleukin (IL)-4 and IL-10 levels. Similarly, in the most recently published RCT on this topic, Wan et al. [104] randomized 76 severe TBI patients into either a control group, receiving enteral nutrition alone, or an intervention group, receiving enteral nutrition in addition to probiotics. At both day 7 and day 15 postintervention, the probiotics group had significantly lower levels of IL-6, IL-10, and tumor necrosis factor (TNF)-α, along with reduced hospital stays and lower rates of respiratory infection. However, Glasgow Coma Scale scores in the probiotics group were lower than those in the control group. In a retrospective cohort study, Painter et al. [105] compared outcomes of TBI patients who had received a standardized nutrition formula compared to TBI patients who received a nutrition formula with higher levels of prebiotics, named an immune-enhancing nutrition (IEN) formula. Results showed that patients who received the IEN formula had lower rates of bacteremia (*p* < 0.05) and significantly higher levels of prealbumin, a potential marker of improved nutrition, in weeks 2 (*p* = 0.006) and 3 (*p* = 0.04) after admission when compared to the standardized nutrition formula control group. However, those in the IEN group had longer ICU stays and higher utilization of ventilators.

Overall, it appears that there is evidence of a positive effect from probiotic use on neurologic injury outcomes. This observation is likely derived from an antiapoptotic effect, including downregulation of Bax and caspase-3 and upregulation of Bcl-2 expression, and other anti-inflammatory mechanisms. It appears that probiotic treatment in the clinical context may be most beneficial in attenuating infection rates and reducing inflammation. Additionally, probiotics are now widely available for use in clinical and nonclinical settings, deemed to be relatively inexpensive, and have been generally demonstrated to be safe for human consumption [106]. However, there is still a relative sparsity of human clinical studies on this subject, and a greater amount of research is needed to be able to determine the safety and efficacy of probiotic/prebiotic intervention more accurately in this patient population before recommendations for clinical use can be made.

### 4.2. Fecal Microbiota Transplantation

In the last decade, fecal microbiota transplantation (FMT) has gained significant interest in relation to gut microbiome interventions on neurologic injury outcomes. However, there have not been any controlled clinical studies evaluating the therapeutic benefits of FMT on human neurologic injury patient outcomes.

Intestinal dysbiosis following stroke and its association with increased inflammatory markers and several poststroke sequelae, such as poststroke cognitive impairment, are now well-characterized [107,108,109]. Regulation of immune cell function seems to play a prominent role in microbiota mediation of stroke pathology. Transplantation of dysbiotic microbiota from poststroke mice into germ-free mice induces a proinflammatory T cell reaction in the gut, and in vivo cell tracking demonstrates that these intestinal lymphocytes can then traffic to the brain [110]. Fecal transplant from healthy mice to poststroke mice is protective against stroke, but not in T-cell-deficient Rag1−/− mice [110], suggesting the lymphocyte–microbiota interaction is critical for mediating the gut–brain axis.

As described earlier, Lee et al. [68] transplanted either donor microbiome from young mice or donor microbiome from aged mice into mice that underwent experimental ischemic stroke 3 days prior. Results demonstrated that the mice that received young donor FMT demonstrated higher levels of poststroke behavioral development and lower levels of cerebral and intestinal inflammation. Further, researchers identified that the young donor microbiome had significantly higher levels of short-chain fatty acids (SCFAs) than the aged donor microbiome, which the authors recognized as the cause of these positive benefits. Similar results have since been achieved in spinal cord injury mice, with FMT facilitating functional recovery and neuron regeneration that was also found to correlate with increased levels of SCFAs in mice that received FMT [111]. Butyric acid has been correlated with the highest level of neuroprotection against ischemic stroke in mouse models [112]. More recently, studies have correlated higher levels of the metabolite trimethylamine-N-oxide (TMAO) derived from intestinal microbiota with larger cerebral infarct size and subsequent increased level of poststroke impairment [113]. A 2021 study [76] demonstrated that FMT administered post-TBI in rats was associated with decreased levels of TMAO in the brain and serum, as well as increased levels of the antioxidant enzyme methionine sulfoxide reductase A (MsrA). It is unclear whether these results would hold true with human subjects, but they nonetheless add credence to the impact that intestinal microbiota may have on the severity of and level of recovery from neurologic injury.

FMT is a novel technique that is beginning to gain traction in the realm of recovery from neurologic injury. Benefits, including increased functional and behavioral recovery in animal models, have been largely attributed to the increased production of SCFAs and the reduction of metabolites such as TMAO. The lack of human studies on the efficacy of this intervention limits its current applicability in the clinical setting.

### 4.3. Oral Short-Chain Fatty Acids (SCFAs)

As increased levels of SCFAs from FMT recipient models have been shown to increase positive outcomes from neurologic injury, researchers have recently begun experimenting with orally administered SCFAs in animal neurologic injury models [114,115,116]. In 2020, Sadler et al. [117] treated mice with 4 weeks of oral SCFA supplementation before implementation of experimental stroke. Compared with controls, mice that received oral SCFA supplementation displayed significantly reduced motor deficits (*p* = 0.01) measured by a lever pull test of the affected limb. Further, they found a higher level of activation of circulating lymphocytes and a subsequent greater degree of microglial activation in SCFA-treated mice, suggesting a potential regenerative effect on neural plasticity as the mechanism by which SCFAs may benefit poststroke recovery. Furthermore, in 2021, Opeyemi et al. [77] randomized 20 experimentally induced TBI mice into a control group, receiving standard drinking water for 2 weeks before TBI, and an intervention group, receiving SCFA supplementation. The SCFA-supplemented group showed greater capacity for spatial learning measured 2 weeks post-TBI using the Morris water maze.

The impact of orally administered SCFAs on neurologic injury outcomes is still being investigated and has not been adequately tested on human neurologic injury patients in a clinical setting. However, SCFAs appear to have a relatively safe profile and can be administered easily through oral intake [118]. Thus, although more research is needed to evaluate efficacy in human subjects in this context, oral SCFAs may eventually serve as a safe and easy-to-use adjunct to post-neurologic injury therapy to improve patient outcomes.

### 4.4. Other Novel Interventions

There have been several other alternative therapies evaluated in the last decade using neurologic injury animal models. Liu et al. [119] demonstrated that administration of the flavonoid baicalin to mice that underwent cerebral ischemia-reperfusion injury reduced levels of TMAO; increased hippocampal density; and improved cognition, memory, and long-term potentiation compared to controls. Furthermore, they noted decreases in these benefits when mice were pretreated with an antibiotic regimen to deplete the intestinal microbiota, suggesting the mechanism by which baicalin exerts its positive effects is through the intestinal microbiome. Zhang et al. [120] used TBI rat models to test the effect of direct intestinal injection of carbon monoxide-releasing molecule (CORM)-3 on several outcomes, including inflammatory cytokine levels and functional outcomes. Results showed that rats treated with CORM-3 demonstrated reduced serum levels of IL-1β and IL-18 levels 24 h post-TBI and increased measures of learning, memory, and exploratory activity. Pang et al. [121] showed that rats orally treated with the plant *Dioscorea polystachya* after cerebral ischemia-reperfusion injury demonstrated heightened levels of brain-derived neurotrophic factor, which authors linked to the subsequently measured increased intestinal levels of SCFAs and probiotic bacteria compared with controls.

To summarize, probiotic/prebiotic administration and FMT are being investigated as potential adjunct therapeutic interventions in aiding recovery from neurologic injury. However, data on these interventions using human subjects in a clinical setting are currently severely limited. Oral SCFA and baicalin administration have shown limited potential in the current state. An overview of the emerging gastrointestinal microbiome-related therapeutic interventions on neurologic injury outcomes is presented in Table 1.

## 5. Conclusions

The communication between the gut and the brain is a complex interplay that is not yet fully understood. There has been an increased emphasis on the role that the gut microbiota plays in this relationship. Neurologic injury can lead to gut dysbiosis leading to challenges in the recovery process. Additionally, gut dysbiosis is a possible contributing factor to various neurologic diseases and injuries. Interventions such as probiotics/prebiotics and fecal microbiota transplant have shown promise in aiding in the recovery process from neurologic injury. Further human clinical trials are needed to understand the clinical advantages and disadvantages these various interventions exhibit.

## Figures and Tables

**Figure 1 biomedicines-10-00500-f001:**
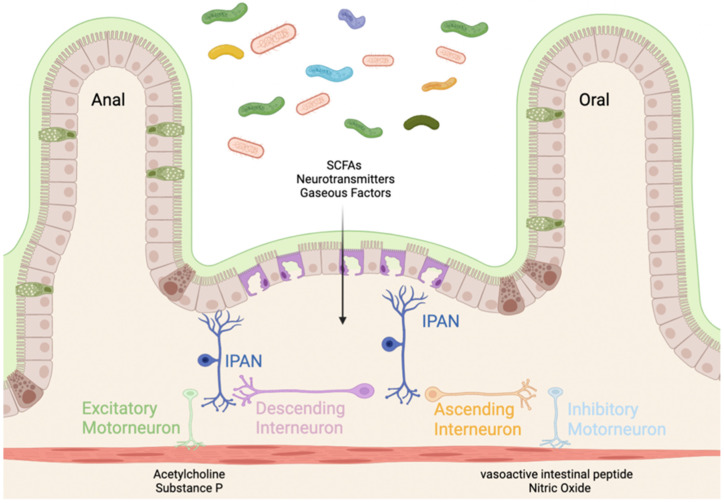
Innervation of the gut through the enteric nervous system (ENS). Mechanical and chemical sensory information is detected by the intrinsic primary afferent neurons (IPANs). The gut microbiota secretes short-chain fatty acids (SCFAs), neurotransmitters, and gaseous factors (e.g., NO). The signal is then transported by ascending (projected orally) and descending (projected anally) interneurons. The signal is transferred to excitatory motor neurons (releasing acetylcholine (Ach) and substance P (SP) to contract the enteric musculature) and inhibitory motor neurons (releasing vasoactive intestinal peptide (VIP) and nitric oxide (NO)) [41]. Created with BioRender.com (accessed on 23 January 2022).

**Table 1 biomedicines-10-00500-t001:** Emerging gastrointestinal microbiome-related therapeutic interventions on neurologic injury outcomes.

Intervention	Type of Research	Main Findings	Advantages	Disadvantages
Probiotics/ prebiotics	Clinical, preclinical (mice)	- Preclinical studies highlight antiapoptotic and anti-inflammatory effects, improved neurologic function - Clinical studies highlight reduced infection rates and inflammatory markers, mixed outcomes on length of hospital stay and long-term outcomes	- Largest research base of the emerging interventions listed - Widespread availability - Cheap - Safe - Orally administered	- Limited generalizability from preclinical data due to inter-species differences in microbiome composition and inclusion of potentially clinically ineffective pretreatment regimens
Fecal microbiota transplant	Preclinical (mice, rats)	- Increased levels of SCFAs - Facilitated functional and behavioral recovery - Decreased gut-derived metabolite TMAO shown to correlate with several negative poststroke outcomes	- Becoming more widely used in the clinical setting for other indications - Does not require daily supplementation	- No intervention-based clinical data on outcomes using humans - More invasive than probiotic/oral SCFA supplementation
Oral short-chain fatty acids	Preclinical (mice)	- Reduced poststroke motor deficits and enhanced post-TBI spatial learning - Activated circulating lymphocytes and resident microglia to induce regeneration of neural plasticity	- Noninvasive - Safe	- No intervention-based clinical data on outcomes using humans - Limited preclinical data compared to probiotics/prebiotics and FMT
Baicalin	Preclinical (mice)	- Reduced levels of TMAO - Increased poststroke hippocampal density - Enhanced poststroke cognitive recovery	- Can be orally administered	- No intervention-based clinical data on outcomes using humans - Data from only one preclinical study
CORM-3	Preclinical (rats)	- Reduced post-TBI inflammatory markers - Enhanced post-TBI cognitive and functional recovery	- Demonstrated positive effects on both cognitive and functional recovery domains	- No intervention-based clinical data on outcomes using humans - Data from only one preclinical study - Requires direct intestinal injection
*Dioscorea polystachya*	Preclinical (rats)	- Increased intestinal SCFA and probiotic levels poststroke - Increased poststroke level of brain-derived neurotrophic factor	- Can be orally administered	- No intervention-based clinical data on outcomes using humans - Data from only one preclinical study

## Data Availability

Not applicable.

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
