# Peer review of "Gastrointestinal Microbiome and Neurologic Injury"

_biomedicines, 2022, doi:10.3390/biomedicines10020500_

Round 1

Reviewer 1 Report

This is a comprehensive, very well-written review that is novel and advances the field of microbiome interactions in disorders of neurological damage. My main request is to clarify which studies are preclinical in rodents and which are in humans. This is done well in some areas but not others (see below). I would also like to see more indication of the quality of the human work as often these early clinical trials have study flaws. There is a lack of critique and synthesis that could make this a stronger review overall. Finally, systematic reviews rather than narrative reviews seem to be preferred in the current climate. Addition of the search and inclusion strategy could also add to the strength of the manuscript.

Specific comments:

  1. I think it would be important to emphasize with the human studies that the microbiome changes could be epiphenomenon of the disease state rather than causative. For example, an unhealthy diet and lifestyle could predispose to both stroke and unfavorable microbiome changes. The preclinical studies are important in this regard to support the causality element. It is also important, however, to acknowledge that what happens in rodents does not necessarily translate to people. I believe it would be helpful to make these  caveats explicit.
  2. Intro to TBI section (3.3): The first few sentence are strongly stated. Are these studies in rodents or humans? Maybe this should be phrased less as fact and more as prior findings?
  3. Line 310: Do you spell out Glasgow Coma Scale somewhere earlier? If not, should spell it out. Non-physicians will not know this abbreviation.
  4. Line 329: Safety in addition to efficacy should be confirmed given the mixed outcome findings in preliminary studies.
  5. The paragraph starting at line 345: Are these the same studies described earlier? I was wondering if it was repetitive and if so, maybe stating "as described earlier"  and that this pertains also to treatment?
  6. The second half of this same paragraph seemed to  discuss conflicting data but it was not spelled out that it is conflicting nor was there any hypotheses for why this might be the case.
  7. Line 372: last word of this line should be "of" and not "on," I believe.
  8. In the table: I'm not sure pretreatment regimens are irrelevant, because this information could inform preventive care for high risk individuals.

Author Response

Please see attachment under "Reviewer 1".

Your comments were very thorough and valuable. Thank you for your comments and we are confident that the revisions have strengthened the manuscript and that it will be of strong interest to the readership. We are looking forward to the page proofs. 

Reviewer 2 Report

Dear Sirs,

this is a very well-written and very easy to read manuscript, albeit the difficulties posed by its theme per se. The authors have succeeded in making this issue more attractive to the readers. My only comment is that the way of research, ie Pubmed using the words.... should be added. In addition, are there any reports for next generation probiotics, such as Akkermansia muciniphila or Enterobacterium prauznitsii regarding neurological disorders? Nevertheless, I recommend publishing this manuscript with these two minor additions.

Author Response

Thank you for your comments. Please see the attachment under "Reviewer 2". 

We are confident that the revisions have greatly strengthened the manuscript and that it will be of strong interest to the readership. We are looking forward to the page proofs.
